

# Characteristics of craniofacial morphology and factors affecting them in patients with isolated cleft palate

Kyoko Tsuji[1],*, Naoto Haruyama[1],*, Shunsuke Nomura[1],
Naohisa Murata[1], Keigo Yoshizaki[1], Takeshi Mitsuyasu[2],
Hiroyuki Nakano[3], Seiji Nakamura[2], Yoshihide Mori[4] and
Ichiro Takahashi[1]

[1] Section of Orthodontics and Dentofacial Orthopedics, Faculty of Dental Science, Kyushu University, Fukuoka, Fukuoka, Japan
[2] Section of Oral and Maxillofacial Oncology, Faculty of Dental Science, Kyushu University, Fukuoka, Fukuoka, Japan
[3] Department of Dentistry and Oral Surgery, Division of Medicine for Function and Morphology of Sensory Organs, Faculty of Medicine, Osaka Medical and Pharmaceutical University, Takatsuki, Osaka, Japan
[4] Section of Oral and Maxillofacial Surgery, Faculty of Dental Science, Kyushu University, Fukuoka, Fukuoka, Japan
* These authors contributed equally to this work.

Corresponding author
Naoto Haruyama,
haruyama@dent.kyushu-u.ac.jp

## ABSTRACT

**Background:** Myriad maxillo-mandibular occlusal relationships are observed in patients with isolated cleft palate (ICP), unlike in patients with other cleft types, such as cleft lip and palate.

**Objectives:** This study aimed to categorise the characteristics of craniofacial morphology in patients with ICP, and investigate the clinical factors affecting these categorised morphological characteristics.

**Methods:** Thirty-six girls with ICP (age (mean ± SD): 5.36 ± 0.36 years) underwent cephalometric measurement. Their craniofacial morphology was categorised using cluster analysis. Profilograms were created and superimposed onto the standard Japanese profilograms to visualise the morphological characteristics of each group (cluster). The mean values and variations in the linear and angular measurements of each group were compared with the Japanese standards and statistically analysed using Dunnett's test after the analysis of variance. Fisher's exact test was used to analyse the differences between the cleft types (cleft in the hard and/or soft palate) and skills of the operating surgeons in the groups.

**Results:** Cluster analysis of craniofacial morphologies in patients with ICP resulted in the formation of three categories: the first cluster exhibited a relatively harmonious anteroposterior relationship between the maxilla and the mandible (22.2%); the second cluster exhibited crossbite owing to a significantly smaller maxilla (33.3%); and the third cluster exhibited a smaller mandible with posterior rotation showing skeletal class II malocclusion (44.4%). Differences in cleft types and surgeons were not associated with the distribution of patients in each cluster.

**Conclusions:** Patients with ICP exhibited characteristic morphological patterns, such as bimaxillary retrusion or severe mandibular retrusion, besides the anterior crossbite frequently found in patients with cleft lip and palate . Understanding the typical

morphological characteristics could enable better diagnostic categorisation of patients with ICP, which may eventually improve orthodontic treatment planning.

## INTRODUCTION

Cleft lip and/or palate (CL/P) is one of the most common congenital anomalies in the orofacial area (*Schutte & Murray, 1999*). Patients with cleft palate including the isolated cleft palate (ICP) and the cleft lip and palate (CLP) generally undergo palatoplasty at the age of 1–1.5 years to obtain velopharyngeal competence and normal speech. The various palatoplasty techniques, such as Von Langenbeck's method (*Von Langenbeck, 1861*), Veau–Wardill–Kilner push-back method (*Wardill, 1937*), intravelar veloplasty (*Braithwaite & Maurice, 1968*) and Furlow double opposing Z-plasty (*Furlow, 1986*), have been introduced by different surgeons. Although there are many variations of these techniques, the scar formed on maxillary oral and nasal mucosa by palatoplasty is thought to impair maxillary growth, which is probably responsible for the high frequency of anterior crossbite in patients operated for cleft palate. For example, a survey of 996 Japanese patients with CLP (*Kouno, Suzuki & Watanabe, 1989*) found that the reverse occlusion involving at least one tooth was observed in 84.6% of CLP cases. *Vettore & Sousa Campos (2011)* also observed that the frequencies of anterior crossbite and posterior crossbite in Brazilian patients with CLP were 60.7% and 39.3%, respectively. *Sæle et al. (2017)* demonstrated that 61.4% of Norwegian patients with CLP had Angle's class III malocclusion.

However, the suppression of maxillary bone growth due to palatoplasty is not necessarily a problem in patients with ICP. Hermann et al. investigated unoperated patients with ICP and suggested that children with ICP had a shorter maxilla, reduced maxillary posterior height, shorter mandible, and reduced posterior height of the mandible with retrognathia, compared to controls with unilateral isolated cleft lip (ICL) (*Hermann et al., 2002*). On the other hand, *Fujita et al. (2005)* reported that the average maxillary length was shorter and the nasomaxillary complex was generally positioned more posteriorly in operated patients with ICP than in controls with normal occlusion; however, the craniofacial pattern exhibited a wide variation in patients operated for ICP. *Nakasone et al. (2013)* observed a variety of mandibular configurations and jaw relationships, indicating the diversity of craniofacial morphologies in patients operated for ICP. These reports imply that the characteristics of craniofacial morphology are not sufficiently understood by simply comparing the cephalometric mean values between patients operated for ICP and controls, such as those with unilateral ICL or the normal population.

The objective of this study was to categorise the characteristics of craniofacial morphology in patients operated for ICP, to elucidate the manifestation patterns before orthodontic treatment, and to investigate the causal relationships between morphological

categories and potential key factors, such as the differences in cleft type (range of cleft) and skills of the operating surgeons.

## MATERIALS & METHODS

### Sample and data collection

This study protocol was reviewed and approved by the Kyushu University Institutional Review Board for Clinical Research (#27-135). A dedicated website was established with detailed information on this clinical study in case patients wished to opt out of the investigation. For this retrospective cohort study, the participants were selected from among patients with ICP who first visited the Department of Orthodontics at Kyushu University Hospital between 2002 and 2014.

The inclusion criterion was: patients with overt ICP (i.e., clefts of the hard and soft palate or clefts of the soft palate) ($n$ = 77). The exclusion criteria were: patients with syndromes affecting the craniofacial morphology including the Pierre Robin sequence ($n$ = 1), patients who underwent palatoplasty other than push-back method ($n$ = 7), missing diagnostic records according to being transferred to other hospitals or the default of their appointment ($n$ = 16), and incomplete diagnostic records (i.e., cephalogram taken with the mouth opened) or unwanted developmental stages (i.e., other than Hellman's dental stage IIA (Completion of the primary dentition by the acquisition of second deciduous molars) (*Hellman, 1935*)) ($n$ = 5). Of the 48 selected patients, 12 boys were excluded from the study sample because their numbers were not sufficient for the categorisation and comparison of craniofacial morphology. Thirty-six girls, who underwent lateral cephalography before orthodontic diagnosis, belonging to Hellman's dental stage IIA were eventually included in the study. The 36 patients were aged 5.36 ± 0.36 years (mean ± SD). The participants were evaluated based on the cleft type (range of cleft): 23 patients had clefts of the hard and soft palate and 13 had clefts of the soft palate. All primary surgeries were performed by a group of similarly trained oral surgeons, and using the modified Wardill's push-back method at 1.55 ± 0.21 years (mean ± SD) of age.

### Cephalometric measurements and cluster analysis

All the cephalograms were taken on the same machine to account for the magnification factor. The magnification factor in lateral cephalometric radiographs was 110% at the midsagittal plane. WinCeph 10.0 (Rise Corp., Sendai, Japan) was used to perform the cephalometric measurements, according to previous studies (*Iizuka, 1958*; *Sakamoto, 1959*), using the reference points represented in Fig. 1. The explanations for all reference points are indicated in the legend.

All cephalometric measurements were conducted by the same orthodontist to eliminate inter-examiner errors. All measurements were recorded again after two months. Dahlberg's formula (*Dahlberg, 1940*) was used to evaluate the reproducibility of the measurements. The measurement error (distance) of the coordinates of 17 points used in the profilogram (*Sakamoto, 1959*) was 0.61 mm. The measurement error (angle) of the 10 representative angular measurements (Facial angle, Convexity, A-B plane, $Y$-axis, FH to SN, SNA, SNB, ANB, N-Pog to SN, and Nasal floor to SN) was 0.43°.
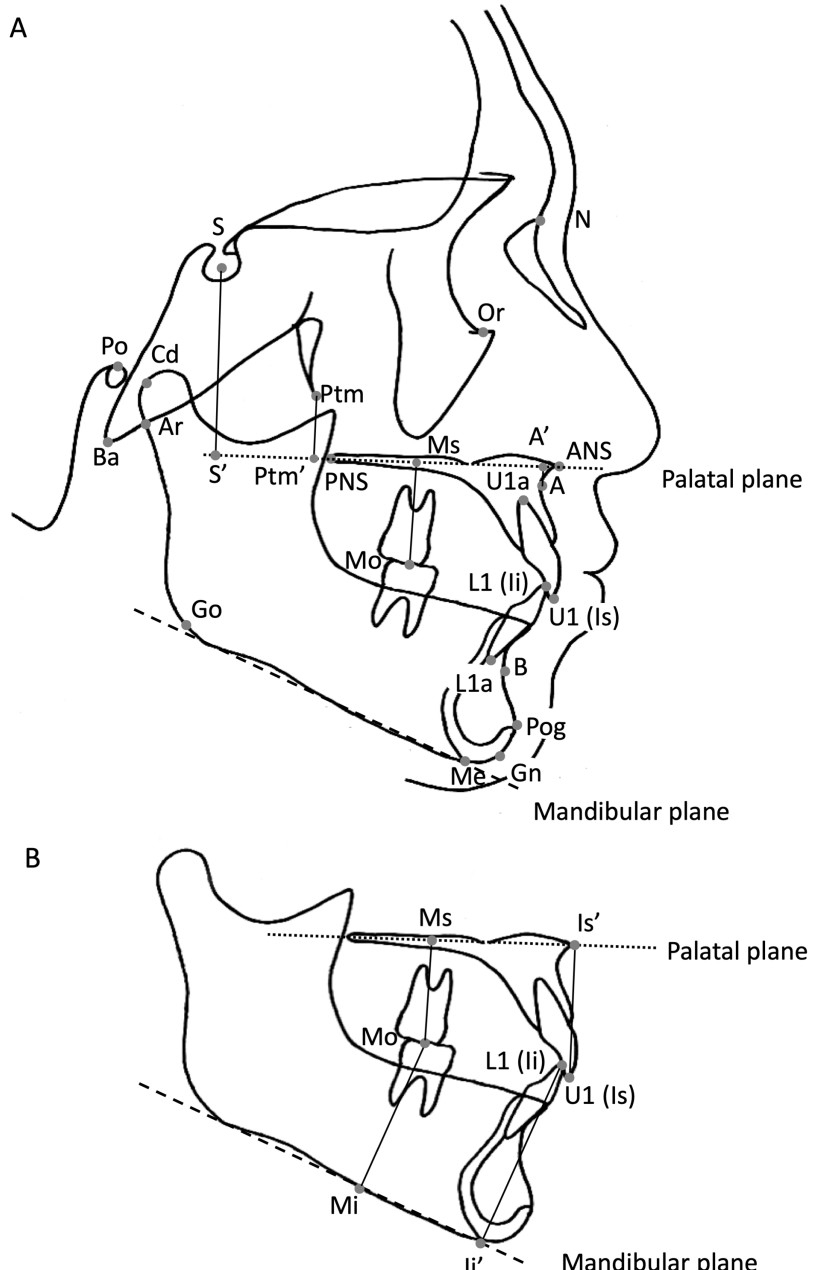

**Figure 1 Reference points for the cephalometric measurements.** (A) N: nasion, the anterior-most point of the frontonasal suture; S: sella turcica, the estimated centre of the hypophyseal fossa; Or: orbitale, the deepest point on the infraorbital margin; Po: porion, the upper margin of the ear canal; Ba: basion, the anterior-most margin of the foramen magnum; Cd; condylion, the most supradorsal point of the condylar head; Ar: articulare, the intersection between the external contour of the cranial base and dorsal contour of the condylar head and neck; Ptm: pterygomaxillary fissure, the inferior point of the fissure; PNS: posterior nasal spine; ANS: anterior nasal spine; A: subspinale, the deepest point on the premaxilla between the ANS and prosthion in the midline; U1 (Is) and U1a : upper incisor constructed between the incisal tip of the anterior-most deciduous maxillary central incisor and its apex; Mo: mid-point of the deciduous maxillary second molar; L1 (Ii) and L1a: lower incisor constructed between the incisal tip of the most anterior deciduous mandibular central incisor and its apex; Pog: pogonion, the anterior-most point of the bony chin; B: supramentale, the posterior-most point in the concavity between the infradentale and Pog; Me: menton, the lowest point on the symphyseal shadow; Gn: gnathion, the point on the

**Figure 1 (continued)**
chin determined by bisecting the angle formed by the facial plane and mandibular plane; Go: gonion, the intersection between the ramus plane and the mandibular plane; A': intersection of a perpendicular drawn from point A to the palatal plane; Ptm': intersection of a perpendicular line from the Ptm to the palatal plane; S': intersection of the perpendicular drawn from S to the palatal plane; palatal plane: the square dotted line; mandibular plane: the dashed line (B) U1 (Is): upper incisor constructed between the incisal tip of the most anteriorly placed deciduous maxillary central incisor; Is': intersection of the perpendicular drawn from Is to the palatal plane; L1 (Ii): lower incisor constructed between the incisal tip of the most anteriorly placed deciduous mandibular central incisor; Ii': intersection of the perpendicular drawn from Ii to the palatal plane; Mo: mid-point of the deciduous maxillary second molar; Ms': intersection of the perpendicular drawn from Mo to the palatal plane; Mi': intersection of the perpendicular drawn from Mo to the mandibular plane; palatal plane: the square dotted line; mandibular plane: the dashed line.           

Cluster analysis was performed to categorise the characteristics of craniofacial morphology using JMP Pro14 (SAS Institute Inc. Cary, NC, USA). The variables used for cluster analysis were based on a previous study (*Yamanouchi et al., 1995*) and included the lower anterior facial height (anterior nasal spine to menton (ANS-Me)), length of the maxilla (point A-pterygomaxillary fissure (A'-Ptm')), length of the mandible (condylion-gonion (Cd-Gn)), anteroposterior position of the maxilla (sella, nasion, point A (SNA)), anteroposterior relationship of the maxilla and mandible (ANB), facial profile (facial angle), mandibular shape (gonial angle), and mandibular rotation (ramus plane to the sella-nasion line (SN)). The linear measurements were normalised based on the upper anterior facial height (nasion to ANS (N-ANS)) before the cluster analysis, to eliminate the effect of the differences in the size of each individual (*Yamanouchi et al., 1995*). A best-suited model of three clusters was obtained using the Ward's method, defining the distances between groups and establishing the least possible dispersion within groups to ensure the greatest homogeneity for each cluster (Fig. 2) (*De Frutos-Valle et al., 2020*).

Average profilograms of each cluster were obtained and superimposed over the standard Japanese age matched female profilogram (*Sakamoto, 1959*) to visualise the characteristics of craniofacial morphology of each cluster. Furthermore, the mean linear or angular measurements of each cluster were statistically compared to the Japanese female standard of linear measurements (*Sakamoto, 1959*) or to the Japanese standard of angular measurements obtained from combined male and female subjects (*Iizuka, 1958*), respectively.

## Statistical analysis

Multiple comparisons were performed using Dunnett's test following an analysis of variance (ANOVA) by Prism 7 (GraphPad Software, San Diego, CA, USA). Moreover, Fisher's exact test was performed to determine whether the differences in cleft type (cleft in the soft palate and/or hard palate) or surgeons (who performed the palatoplasty) affected the craniofacial morphology in each cluster by R (the R Foundation, https://www.r-project.org). Post-hoc power $(1-\beta)$ analysis for the ANOVA and Fisher's exact test was performed by G*power 3.1 (*Faul et al., 2009*). $P < 0.05$ was considered statistically significant.

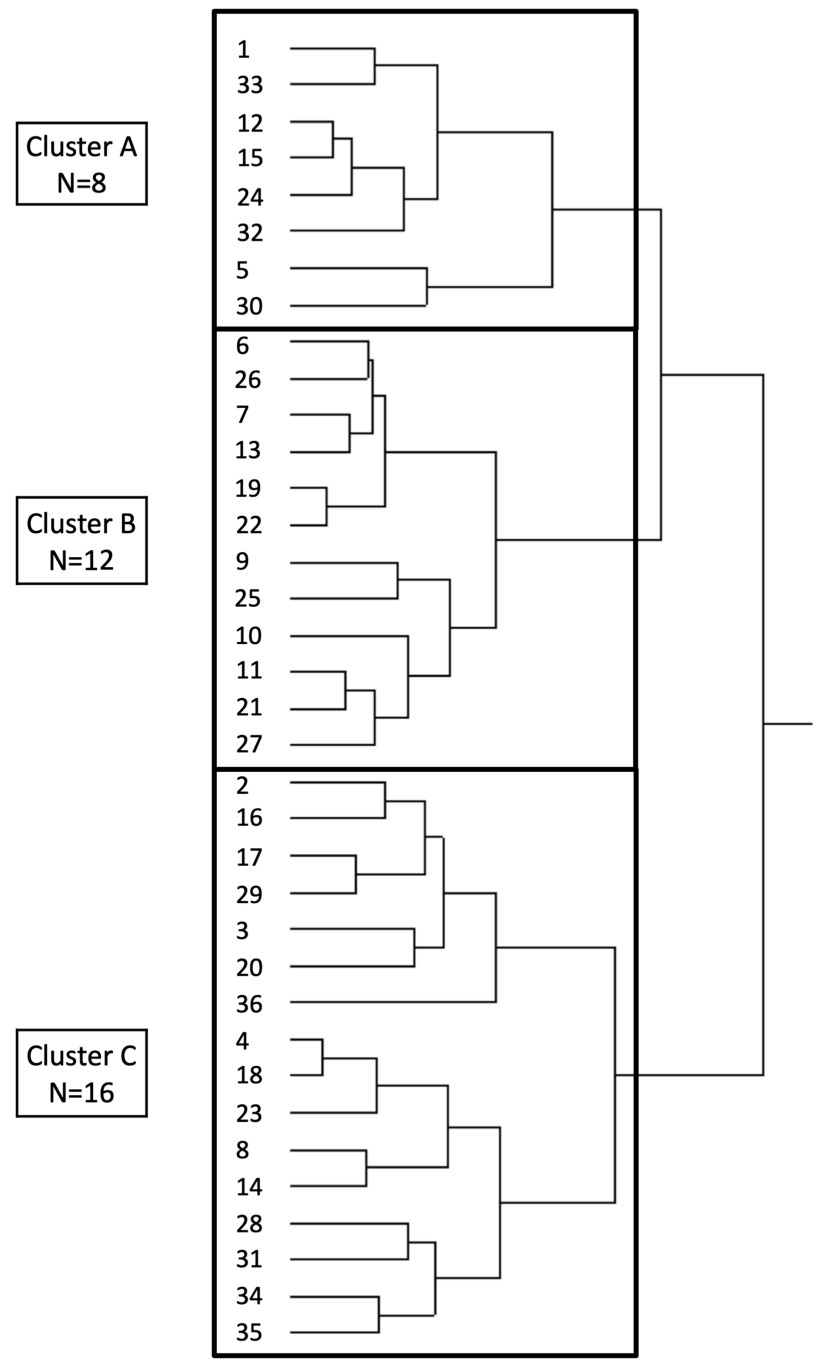

**Figure 2 Dendrogram created by cluster analysis using Ward's method.** Thirty-six girls were classified into three groups, i.e. clusters A, B, and C.

## RESULTS

### Qualitative assessment of craniofacial morphology of each cluster categorised by cluster analysis

Participants were categorised into the following three clusters based on the results of the cluster analysis: cluster A ($n = 8$), cluster B ($n = 12$) and cluster C ($n = 16$) (Fig. 2).

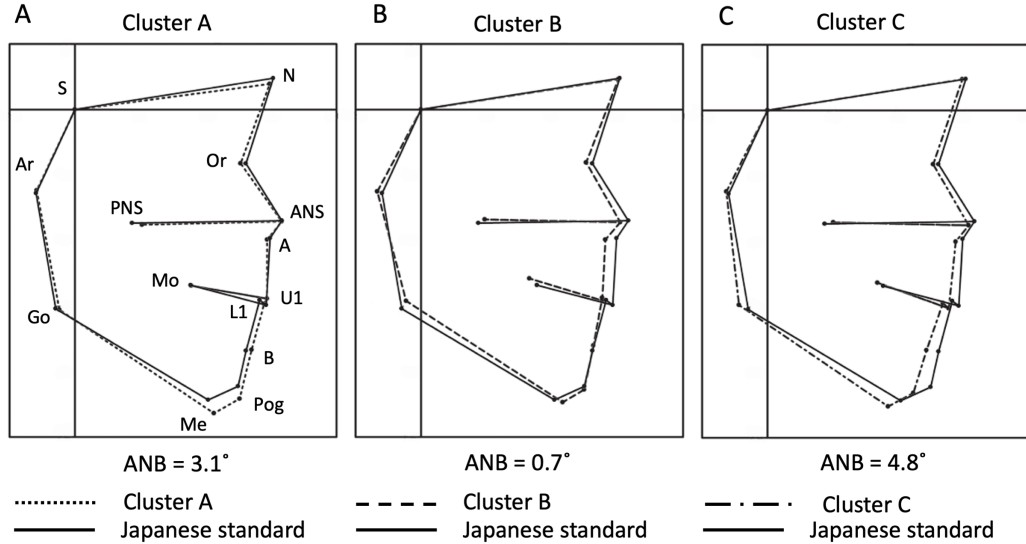

**Figure 3 Averaged profilograms of each cluster and their superimposition over the Japanese standard profilogram.** Points S, N, Or, ANS, A, U1, L1, B, Pog, Me, Go, Ar, PNS and Mo were connected on the profilograms to visualise the facial pattern (according to a previous study (*Sakamoto, 1959*)). The superimposition was performed at point S, parallel to the Frankfurt horizontal (FH) plane. The square dotted line, dashed line, dash-dotted line, and solid line represent clusters A, B, C, and the Japanese standard, respectively. (A) Cluster A exhibited no difference in maxillary position compared to the standard. The mandible and maxilla showed a balanced anteroposterior relationship. (B) Cluster B exhibited significant maxillary retrusion and tendencies towards lingual inclination of the lower incisors, a shorter mandibular ramus, and a larger gonial angle, with crossbite. (C) Cluster C exhibited mandibular and maxillary retrusion, a larger ramus angle, and clockwise rotation of the mandible. The ANB in the averaged profilograms of clusters A, B, and C was +3.1°, +0.7° and +4.8°, respectively. Abscissa: FH-parallel line through S; Ordinate: FH-perpendicular line through S. ANS, anterior nasal spine; A, subspinale; S, sella turcica; N, nasion; Or, orbitale; U1, incisor tip of the most anteriorly placed deciduous maxillary central incisor; L1, lower incisor tip of the most anteriorly placed deciduous mandibular central incisor; B, supramentale; Pog, pogonion; Me, menton; Go, gonion; Ar, articulare; PNS, posterior nasal spine; Mo, mid-point of the deciduous maxillary second molar.

Cluster A exhibited no difference in the maxillary position compared to the standard, when the average profilograms of each cluster were superimposed over the standard profilogram; however, the mandibular body and anterior facial height were larger than those in the standard profilogram. The anteroposterior relationship of the maxilla and mandible was harmonious in cluster A (Fig. 3A). Cluster B exhibited significant maxillary retrusion and a tendency towards lingual inclination of the lower incisors, a shorter mandibular ramus, and a larger gonial angle than those of the standard. Anterior crossbite was observed (Fig. 3B). Cluster C exhibited mandibular and maxillary retrusion, large ramus inclination, and clockwise rotation of the mandible (Fig. 3C).

Superimposition of the profilograms of all clusters over the standard profilogram revealed that the anterior-most maxillary position was observed the most in the standard, followed by that in clusters A, C, and B, while the most anterior mandibular position was observed the most in cluster A, followed by that in the standard, cluster B, and cluster C (Fig. 4).

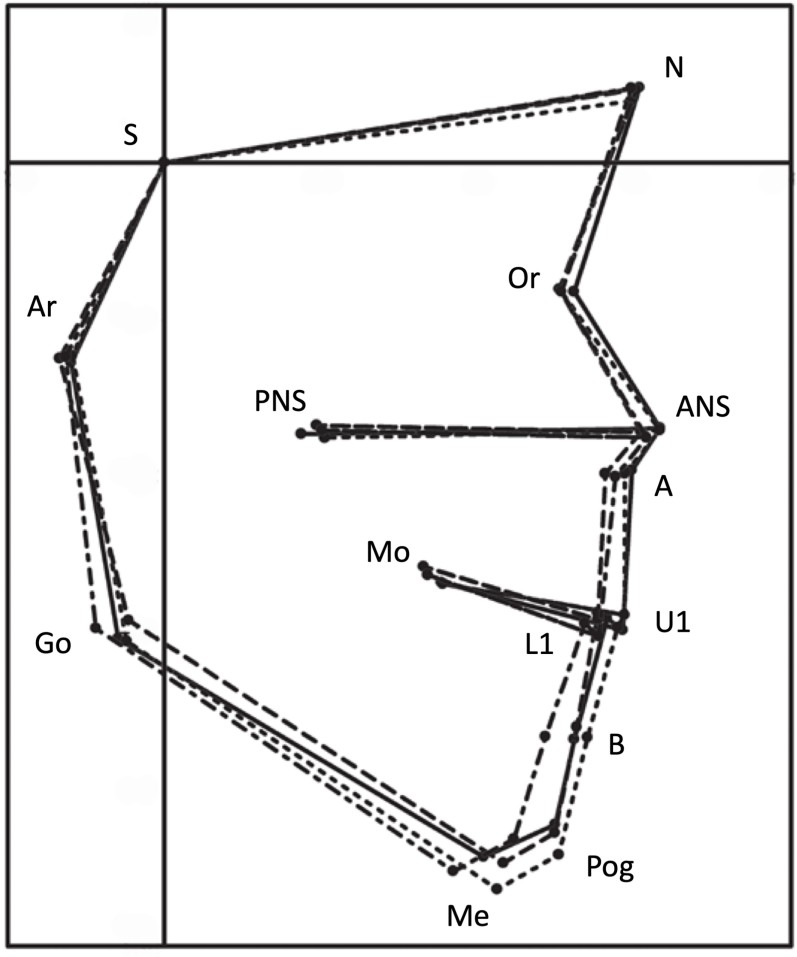

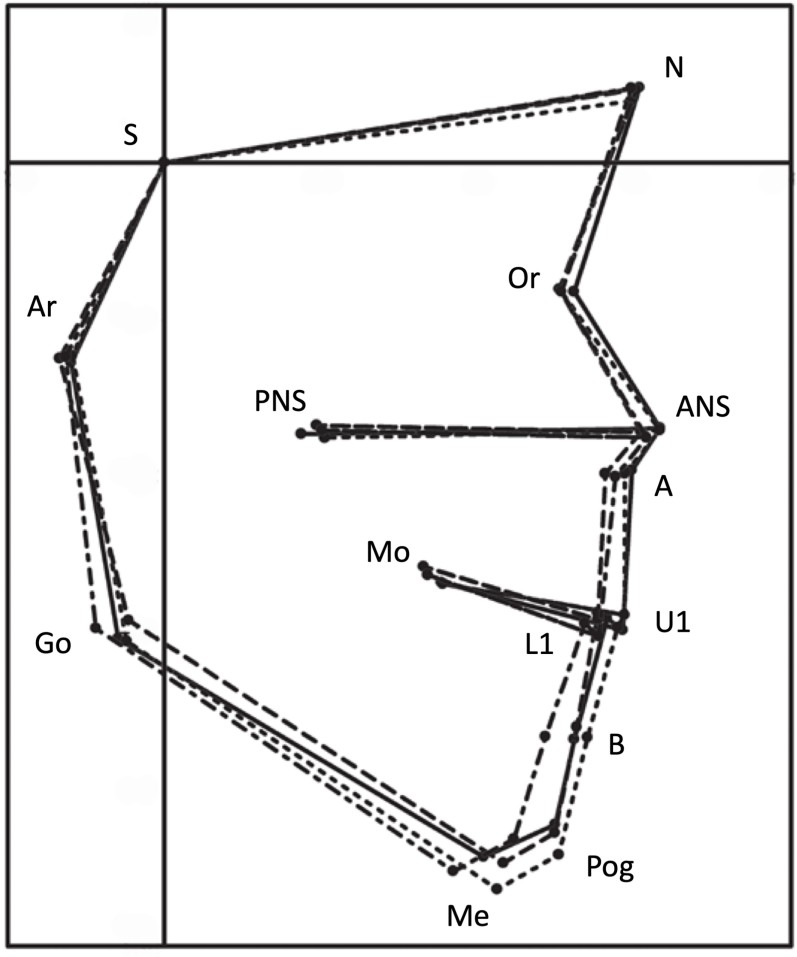

|  |  |
| --- | --- |
| ············· | Cluster A |
| - - - - | Cluster B |
| — · — · | Cluster C |
| ——— | Japanese standard |

**Figure 4 Superimposition of all the clusters over the Japanese standard profilogram.** The super-imposition was performed at point S, parallel to the Frankfurt horizontal (FH) plane. The square dotted line, dashed line, dash-dotted line, and solid line represent clusters A, B, C, and the Japanese standard, respectively. Abscissa: FH-parallel line through S; Ordinate: FH-perpendicular line through S. S: sella turcica.

## Quantitative assessment of facial morphology in each cluster

The mean linear and angular measurements of each cluster were compared to their standard counterparts (*Iizuka, 1958*; *Sakamoto, 1959*) to clarify the morphological characteristics of each cluster (Tables 1 and 2). Cluster A exhibited a significantly longer Gn-Cd (mandibular length) and larger A-B plane angle than the standard, and significantly smaller FH to SN and L1 to mandibular plane angles. These results indicate that the characteristics of cluster A included a larger mandible and slight lingual inclination of the lower incisors. Cluster B exhibited a significantly larger A-B plane and

**Table 1 Comparison between the mean angular measurements of the Japanese standard and each cluster.**

| Angular measurements (°) | Japanese standard | | Cluster A | | | Cluster B | | | Cluster C | | | |
|---|---|---|---|---|---|---|---|---|---|---|---|---|
| | Mean (±SD) | 95% CI | Mean (±SD) | 95% CI | P-value | Mean (±SD) | 95% CI | P-value | Mean (±SD) | 95% CI | P-value | Post-hoc power (1-β) |
| Facial angle | 84.5 (±3.2) | [83.3–85.7] | 84.5 (±1.3) | [83.5–85.6] | 1.000 | 83.7 (±1.3) | [82.9–84.5] | 0.737 | 81.8 (±3.0) | [80.2–83.4] | 0.005** | 0.620 |
| Convexity | 10.6 (±2.7) | [9.6–11.6] | 8.9 (±5.9) | [4.0–13.9] | 0.664 | 3.3 (±5.9) | [−0.5 to 7.0] | 0.000** | 11.9 (±4.8) | [9.3–14.5] | 0.656 | 1.000 |
| A-B plane | −5.7 (±2.4) | [−6.6 to −4.8] | −2.7 (±2.7) | [−5.0 to −0.5] | 0.032* | −0.1 (±4.1) | [−2.7 to 2.5] | 0.000** | −5.9 (±2.7) | [−7.4 to −4.4] | 0.993 | 1.000 |
| Y-axis | 61.5 (±3.4) | [60.2–62.8] | 62.9 (±1.6) | [61.5–64.2] | 0.548 | 62.1 (±1.9) | [60.9–63.3] | 0.906 | 64.6 (±3.5) | [62.7–66.4] | 0.006** | 0.674 |
| FH to SN | 10.1 (±3.0) | [9.0–11.2] | 7.5 (±1.3) | [6.4–8.6] | 0.039* | 8.7 (±2.1) | [7.4–10.1] | 0.300 | 9.5 (±2.5) | [8.1–10.8] | 0.760 | 0.450 |
| ∠SNA | 80.1 (±3.4) | [78.8–81.4] | 81.6 (±3.5) | [78.7–84.5] | 0.502 | 76.6 (±2.6) | [74.9–78.3] | 0.005** | 78.1 (±2.7) | [76.7–79.6] | 0.121 | 0.890 |
| ∠SNB | 76.0 (±3.5) | [74.7–77.3] | 78.5 (±1.5) | [77.3–79.8] | 0.091 | 75.9 (±1.9) | [74.7–77.1] | 0.999 | 73.3 (±2.8) | [71.8–74.8] | 0.012* | 0.842 |
| ∠ANB | 4.1 (±3.5) | [2.8–5.4] | 3.1 (±2.3) | [1.2–5.0] | 0.716 | 0.7 (±2.6) | [−0.9 to 2.3] | 0.003** | 4.8 (±2.1) | [3.7–5.9] | 0.754 | 0.769 |
| N-Pog to SN | 75.3 (±3.8) | [73.9–76.8] | 77.0 (±1.0) | [76.2–77.8] | 0.348 | 75.0 (±1.7) | [73.9–76.1] | 0.977 | 72.3 (±2.6) | [70.9–73.7] | 0.006** | 0.736 |
| Nasal floor to SN | 7.7 (±3.6) | [6.3–9.1] | 6.2 (±2.1) | [4.4–7.9] | 0.412 | 10.0 (±1.9) | [8.8–11.2] | 0.061 | 10.8 (±2.2) | [9.7–12.0] | 0.003** | 0.850 |
| Nasal floor to FH | −1.0 (±3.5) | [−2.3 to 0.3] | −1.3 (±2.7) | [−3.6 to 0.9] | 0.988 | 1.3 (±2.4) | [−0.2 to 2.8] | 0.083 | 1.4 (±2.6) | [0.0–2.8] | 0.035* | 0.604 |
| Mandibular plane to SN | 39.2 (±4.4) | [37.5–40.9] | 39.9 (±3.1) | [37.4–42.5] | 0.956 | 40.4 (±4.4) | [37.6–43.1] | 0.785 | 42.3 (±5.0) | [39.6–45.0] | 0.073 | 0.430 |
| Mandibular plane to FH | 29.5 (±3.4) | [28.2–30.8] | 32.4 (±3.2) | [29.7–35.1] | 0.230 | 31.6 (±4.4) | [28.8–34.4] | 0.353 | 32.9 (±6.0) | [29.7–36.1] | 0.041* | 0.825 |
| Ramus plane to SN | 89.2 (±5.5) | [87.1–91.3] | 89.0 (±4.6) | [85.2–92.9] | 0.999 | 86.0 (±2.2) | [84.6–87.4] | 0.104 | 94.9 (±3.3) | [93.1–96.6] | 0.000** | 0.965 |
| Ramus plane to FH | 79.9 (±5.2) | [77.9–81.9] | 81.5 (±4.3) | [77.9–85.2] | 0.673 | 77.3 (±2.9) | [75.5–79.1] | 0.201 | 85.4 (±3.5) | [83.6–87.2] | 0.000** | 0.956 |
| Gonial angle | 130.0 (±5.3) | [128.0–132.0] | 130.9 (±7.3) | [124.8–137.0] | 0.963 | 134.3 (±4.8) | [131.3–137.4] | 0.081 | 127.5 (±6.3) | [124.1–130.8] | 0.355 | 0.795 |
| U1 to SN | 87.2 (±6.5) | [84.7–89.7] | 92.0 (±7.7) | [85.6–98.4] | 0.209 | 89.1 (±6.2) | [85.2–93.1] | 0.759 | 85.2 (±7.8) | [81.0–89.3] | 0.672 | 0.532 |
| U1 to FH | 96.6 (±6.5) | [94.1–99.1] | 99.5 (±6.9) | [93.8–105.2] | 0.592 | 97.9 (±6.6) | [93.7–102.0] | 0.908 | 94.6 (±7.7) | [90.5–98.7] | 0.677 | 0.304 |
| L1 to mandibular plane | 85.7 (±4.1) | [84.1–87.3] | 78.9 (±4.4) | [75.2–82.6] | 0.041* | 78.2 (±7.5) | [73.4–83.0] | 0.006** | 82.0 (±10.4) | [76.4–87.5] | 0.200 | 1.000 |
| Interincisal angle | 148.4 (±9.5) | [144.8–152] | 149.2 (±11.2) | [139.9–158.6] | 0.994 | 152.3 (±7.9) | [147.3–157.3] | 0.538 | 150.5 (±11.8) | [144.3–156.8] | 0.837 | 0.151 |
| Occlusal plane to SN | 22.3 (±3.9) | [20.8–23.8] | 21.6 (±2.1) | [19.9–23.3] | 0.914 | 24.0 (±3.0) | [22.1–25.9] | 0.342 | 26.9 (±3.1) | [25.2–28.6] | 0.000** | 0.932 |
| Occlusal plane to FH | 12.6 (±3.0) | [11.5–13.7] | 14.1 (±1.9) | [12.5–15.7] | 0.490 | 15.2 (±3.2) | [13.2–17.3] | 0.044* | 17.5 (±3.7) | [15.5–19.4] | 0.000** | 0.996 |

**Notes:**
* $P < 0.05$ by Dunnett's multiple comparison test.
** $P < 0.01$ by Dunnett's multiple comparison test.
SD, standard deviation; 95% CI, 95% Confidence Interval.

**Table 2  Comparison between the mean linear measurements of the Japanese standard and each cluster.**

| Linear measurements (mm) | Japanese standard | | Cluster A | | | Cluster B | | | Cluster C | | | Post-hoc power (1-β) |
|---|---|---|---|---|---|---|---|---|---|---|---|---|
| | Mean (±SD) | 95% CI | Mean (±SD) | 95% CI | P-value | Mean (±SD) | 95% CI | P-value | Mean (±SD) | 95% CI | P-value | |
| N-S | 61.5 (±2.3) | [60.2–62.8] | 60.2 (±3.5) | [57.3–63.1] | 0.515 | 61.0 (±1.8) | [59.9–62.2] | 0.940 | 60.6 (±2.6) | [59.2–62.0] | 0.673 | 0.245 |
| N-ANS | 43.7 (±2.9) | [42.0–45.4] | 42.2 (±2.0) | [40.6–43.9] | 0.344 | 43.7 (±1.3) | [42.9–44.6] | 1.000 | 44.8 (±2.4) | [43.5–46.0] | 0.440 | 0.368 |
| ANS-Me | 58.9 (±3.3) | [57.0–60.8] | 62.2 (±4.0) | [58.9–65.6] | 0.127 | 57.8 (±3.5) | [55.6–60.1] | 0.813 | 60.2 (±4.2) | [57.9–62.4] | 0.689 | 0.739 |
| N-Me | 100.0 (±4.2) | [97.6–102.4] | 102.1 (±5.2) | [97.7–106.4] | 0.629 | 100.2 (±4.3) | [97.5–102.9] | 0.999 | 102.3 (±5.1) | [99.6–105.0] | 0.407 | 0.345 |
| S'-Ptm' | 16.9 (±2.1) | [15.7–18.1] | 16.8 (±2.5) | [14.7–18.9] | 0.998 | 17.4 (±1.4) | [16.5–18.3] | 0.823 | 17.6 (±1.5) | [16.8–18.4] | 0.593 | 0.149 |
| A'-Ptm' | 41.9 (±2.1) | [40.7–43.1] | 41.1 (±2.5) | [39.1–43.2] | 0.798 | 39.8 (±1.9) | [38.6–41.0] | 0.061 | 41.0 (±2.6) | [39.6–42.4] | 0.573 | 0.665 |
| Ptm'-Ms | 17.6 (±2.1) | [16.4–18.8] | 17.6 (±2.8) | [15.3–19.9] | 1.000 | 16.8 (±1.2) | [16.1–17.6] | 0.623 | 17.7 (±1.8) | [16.8–18.7] | 0.995 | 0.150 |
| A'-Ms | 24.3 (±1.3) | [23.6–25.1] | 23.6 (±1.4) | [22.4–24.8] | 0.739 | 23.0 (±2.3) | [21.5–24.4] | 0.208 | 23.3 (±2.3) | [22.0–24.5] | 0.352 | 0.783 |
| Is-Is' | 25.5 (±1.8) | [24.5–26.5] | 25.7 (±2.0) | [24.0–27.4] | 0.990 | 24.1 (±2.3) | [22.6–25.6] | 0.161 | 25.1 (±1.5) | [24.3–25.9] | 0.903 | 0.485 |
| Mo-Ms | 18.6 (±1.7) | [17.6–19.6] | 19.0 (±1.9) | [17.4–20.6] | 0.887 | 17.8 (±1.2) | [17.0–18.6] | 0.376 | 18.1 (±1.2) | [17.5–18.8] | 0.702 | 0.289 |
| Is-Mo | 24.0 (±1.3) | [23.3–24.8] | 23.6 (±2.6) | [21.4–25.8] | 0.942 | 23.3 (±2.3) | [21.8–24.8] | 0.734 | 23.0 (±2.1) | [21.9–24.1] | 0.417 | 0.543 |
| Gn-Cd | 89.7 (±3.6) | [87.6–91.8] | 94.5 (±5.4) | [89.9–99.0] | 0.028* | 93.2 (±4.0) | [90.6–95.7] | 0.089 | 90.3 (±3.7) | [88.4–92.3] | 0.956 | 0.924 |
| Pog'-Go | 59.3 (±3.0) | [57.6–61.0] | 61.3 (±3.2) | [58.6–63.9] | 0.309 | 60.7 (±2.4) | [59.2–62.2] | 0.486 | 59.6 (±3.1) | [58.0–61.3] | 0.980 | 0.331 |
| Cd-Go | 44.2 (±3.0) | [42.5–45.9] | 44.5 (±2.8) | [42.2–46.9] | 0.989 | 42.2 (±2.7) | [40.4–43.9] | 0.176 | 42.3 (±2.8) | [40.8–43.8] | 0.184 | 0.573 |
| Ii-Ii' | 34.2 (±1.6) | [33.3–35.1] | 36.5 (±1.9) | [34.8–38.1] | 0.064 | 33.5 (±2.3) | [32.0–35.0] | 0.767 | 35.3 (±2.7) | [33.9–36.7] | 0.404 | 0.978 |
| Mo-Mi | 27.9 (±2.1) | [26.7–29.1] | 29.0 (±1.5) | [27.8–30.3] | 0.499 | 26.8 (±1.9) | [25.7–28.0] | 0.456 | 29.0 (±2.6) | [27.7–30.4] | 0.347 | 0.695 |
| Ii-Mo | 21.1 (±1.3) | [20.4–21.9] | 21.9 (±1.8) | [20.5–23.4] | 0.502 | 21.8 (±2.1) | [20.4–23.1] | 0.603 | 21.1 (±1.3) | [20.4–21.8] | 1.000 | 0.378 |

**Notes:**
* $P < 0.05$ by Dunnett's multiple comparison test.
SD, standard deviation; 95% CI, 95% Confidence Interval.

**Table 3  Relationships of craniofacial morphology with the differences in cleft type and surgeons.**

**(A) Patient distribution according to the cleft type**

|  | Cleft in hard & soft palate | Cleft in soft palate | Total |
|---|---|---|---|
| Cluster A | 3 | 5 | 8 |
| Cluster B | 9 | 3 | 12 |
| Cluster C | 11 | 5 | 16 |
| Total | 23 | 13 | 36 |

**Note:**
$P(\chi^2) = 0.27$, effect size; $w = 0.457$, post-hoc power; $(1-\beta) = 0.725$.

**(B) Patients distribution according to the operating surgeon**

|  | Surgeon A | Surgeon B | Surgeon C | Surgeon D | Total |
|---|---|---|---|---|---|
| Cluster A | 3 | 1 | 1 | 3 | 8 |
| Cluster B | 4 | 2 | 4 | 2 | 12 |
| Cluster C | 5 | 6 | 2 | 3 | 16 |
| Total | 12 | 9 | 7 | 8 | 36 |

**Note:**
$P(\chi^2) = 0.69$, effect size; $w = 0.775$, post-hoc power; $(1-\beta) = 0.957$.

occlusal plane to FH, and significantly smaller SNA, L1 to mandibular plane, convexity, ANB, and anterior incisal height (Is-Is') than the standard. These results indicate that the characteristics of cluster B included a tendency towards a skeletal class III relationship with maxillary retrusion. Cluster C exhibited a significantly larger Y-axis, nasal floor to SN, nasal floor to FH, mandibular plane to FH, ramus to SN, ramus to FH, occlusal plane to SN, and occlusal plane to FH, and a significantly smaller facial angle, SNB, and N-Pog to SN than the standard. These results indicate that the characteristics of cluster C included clockwise rotation of the occlusal unit, including the maxilla and mandible, resulting in a profile exhibiting mandibular retrusion with a skeletal class II relationship.

## Relationships between craniofacial morphology and the differences in cleft type and surgeons

Patients with ICP in this study included those with clefts of the hard and soft palate and cleft of the soft palate only, and were operated on by four different surgeons (surgeons A-D). Thus, the respective relationships of craniofacial morphology with differences in cleft type and surgeons were also analysed. The results of Fisher's exact test for cleft type ($P(\chi^2) = 0.27$) indicated no bias in patients' distribution. The results of Fisher's exact test for surgeons ($P(\chi^2) = 0.69$) also did not indicate any bias. Taken together, these findings indicate that the morphological differences in ICP were not significantly affected by the cleft type or skills of the operating surgeon (Table 3).

## DISCUSSION

Patients with ICP, unlike those with other cleft types (e.g. CLP), present with a wide variety of maxillo-mandibular occlusal relationships (*Fujita et al., 2005*; *Nakasone et al., 2013*). Understanding the typical morphological features is crucial for the correct orthodontic diagnosis and treatment planning. Cluster analysis of craniofacial morphologies in patients

with ICP in the present study resulted in their categorisation into three groups: a cluster with a relatively harmonious anteroposterior relationship between the maxilla and the mandible (22.2%), a cluster with crossbite caused by a significantly smaller maxilla (33.3%), and a cluster with a smaller mandible with posterior rotation showing skeletal class II relationship (44.4%). These results suggest a significant difference from the craniofacial morphology of patients with CLP.

We selected cluster analysis for categorising craniofacial morphologies in this study. Cluster analysis is a method of collecting and classifying items with similar characteristics from a group containing a mixture of differing characteristics (*Finkelstein, Lavelle & Hassard, 1989*). Cluster analysis was considered more suitable for categorising characteristics compared to other methods of analysis, which require prior information about the potential groups, owing to the diversity of craniofacial morphology in patients with ICP.

Several previous studies have used the principal component analysis to determine the variates to be used before conducting the cluster analysis of craniofacial morphologies (*Bui et al., 2006*; *Uribe et al., 2013*; *De Frutos-Valle et al., 2020*). However, it is necessary to understand the clinical significance of the principal components, and sometimes it is even difficult to understand the significance of obtained principal components. Therefore, the variates used for cluster analysis in this study were based on Yamanouchi et al.'s study, which reported that the characteristics of facial morphology were well characterised by linear measurements of the lower anterior facial height (ANS-Me) and lengths of the maxilla and mandible (A'-Ptm' and Cd-Gn) and angular measurements including the anteroposterior position of the maxilla (SNA), anteroposterior relationship of the maxilla and mandible (ANB), facial profile (facial angle), mandibular shape (gonial angle), and mandibular rotation (ramus to SN) (*Yamanouchi et al., 1995*). They also reported that these skeletal factors were not significantly affected by local dental factors.

Soma reported that the differences in the absolute sizes of the jaw or face makes it difficult to compare craniofacial patterns simultaneously and that isometric processing (normalisation of linear measurements) is an effective method for the analysis of craniofacial patterns (*Soma, 1977*). In the present study, normalisation of linear measurements was achieved by dividing the values with the anterior facial height (N-ANS), which is one of the most reliable measurements. This normalization is important to eliminate the effect of individual variation in absolute length on the cluster analysis (i.e., a larger individual may have a greater absolute length compared to a smaller individual when, in fact, the length may be smaller if normalized for size). We believe that the normalisation of linear measurements successfully eliminated the influence of size and better clarified the trends in the characteristics of each sample.

Determination of the number of clusters is subjective and can result in variability between studies. We deemed it appropriate to classify the participants into three cluster groups, which could be understood easily without over-segmentation, based on the dendrogram created by cluster analysis using the Ward's method (Fig. 2). The comparison of the mean values of each cluster (obtained from cephalometric analysis) with the standard values showed that the large mandibular size in cluster A was compensated (dentally) by a slight lingual inclination of the lower incisors. Cluster B had a retruded

maxilla and lingual inclination of the lower incisors, which were similar to the dental compensation observed in cluster A. Cluster C had a retruded mandible with clockwise rotation of the palatal plane, occlusal plane, mandibular plane, and ramus plane. These results, based on cephalometric analyses, were faithfully represented in the superimposed profilograms. Clusters A, B and C constituted 22.2%, 33.3% and 44.4% of the total study population, respectively, suggesting that several patients with ICP have mandibular hypoplasia (retruded mandible; cluster C), in whom the suppression of maxillary growth post-palatoplasty would not lead to major disharmony in the anteroposterior relationship of the jaw. We speculated that the craniofacial morphology in patients with ICP may have formed congenitally or acquired by mechanisms that differ from those in patients with CLP because the craniofacial morphological characteristics of patients with ICP differ from those of patients with CLP, who often exhibit anterior crossbite (*Kouno, Suzuki & Watanabe, 1989*; *Vettore & Sousa Campos, 2011*; *Sæle et al., 2017*).

Several studies that compared ICP with ICL (as the control group) found that patients with ICP had smaller mandibles than those of the controls (*Hermann et al., 2002*; *Eriksen et al., 2006*), which strongly indicates a link between mandibular hypoplasia occurring during mandibular development and ICP. Price et al. reviewed 930 papers including clinical and basic research on the relationship between cleft palate and mandibular hypoplasia, and concluded that mandibular hypoplasia triggers cleft palate formation (*Price, Haddad & Fakhouri, 2016*). Pierre Robin sequence is characterised by a spectrum of anatomical anomalies including mandibular hypoplasia, glossoptosis, life-threatening obstructive apnoea, and feeding difficulties. Typically, a wide U-shaped cleft palate is associated with Pierre Robin sequence (*Lehman, Fishman & Neiman, 1995*); however, a narrow V-shaped cleft is associated with other cleft types. From this perspective, there could be two important causes of ICP: primary failure of palatal fusion (similar to other cleft types), and secondary failure of palatal fusion owing to a small mandible as seen in Pierre Robin sequence. A previous study suggested that the width of the cleft at the posterior end of the hard palate and the total length of the cleft are significantly related to the severity of Pierre Robin sequence (*Godbout et al., 2014*). Unfortunately, the cleft shape or width before palatal closure could not be evaluated in our study because our surgical records were not tailored for the purpose of this study. Therefore, the relationships between the cleft shape/width before surgery and craniofacial morphology could not be determined. Indeed, we did not find any patients with typical manifestations of Pierre Robin sequence such as glossoptosis, life-threatening obstructive apnoea, and feeding difficulties in their earliest infancy in this study. However, patients with mild Pierre Robin sequence cannot be completely distinguished from those with ICP with a small mandible. Therefore, it is conceivable that the number of participants with a relatively wide U-shaped cleft by mild Pierre Robin sequence may have affected the distribution of patients in the three categorised groups.

In a previous study, compensatory growth of the mandible was not observed at least during the first 2 years of life in patients with Pierre Robin sequence (*Hermann et al.,*
*2003*). Moreover, none of the patients with Pierre Robin sequence exhibited significant improvements in the skeletal pattern that could be construed as a gradual correction of the initial severe skeletal class II relationship after the age of 5 years (*Daskalogiannakis, Ross & Tompson, 2001*). These studies also support the notion that this sampling age would be suitable for identifying the typical morphological characters of ICP, even if a certain number of patients with mild Pierre Robin sequence are included in the study.

Nakasone et al. found that while the average severity of maxillary growth suppression was mild in patients with ICP, individual cases displayed a wide variety of facial morphologies, from mandibular protrusion to maxillary protrusion, and that averaging the characteristics of several cases resulted in values that resembled the standard value (*Nakasone et al., 2013*). They also indicated that although the mean value of measurements related to the mandible was similar to the standard value, a large variation was found in mandibular measurements, suggesting the existence of a wide variety of mandibular shapes or positions and maxillo-mandibular relationships in patients with ICP. Therefore, although patients with ICP often present with mandibular hypoplasia at birth, they can present a wide variety of phenotypes, possibly due to the additive effect of the acquired influence of palatoplasty.

One of the limitations of this report is that only girls were selected for the present study. Previous studies have shown that ICP is more common in girls than in boys (*Mossey et al., 2009*; *Martelli et al., 2012*). Similarly, fewer boys with ICP were found than girls with this condition between 2002 and 2014 during the study selection process, i.e. 12 boys and 36 girls. Therefore, boys were excluded from the study sample because their numbers were not sufficient for the categorisation and comparison of craniofacial morphology. Moreover, because of the gender difference in incidence in the ICP, there might be gender-specific aetiology that would affect craniofacial morphology of ICP, suggesting that a group consisting both genders would be inappropriate to be analysed. It will be necessary to examine whether craniofacial morphologies of boys with ICP are in line with the results of the present study.

Based on the appropriate statistical approach, we believe that the results obtained are clinically useful for understanding the typical morphological characteristics of ICP. However, we admit that the retrospective cohort with a relatively small sample may slightly undermine the results. Further evaluation of the prospective cohort study with a large sample would be warranted.

A recent systematic review of the effects of functional appliances suggested that the skeletal changes elicited by orthodontic treatment may be negligible or statistically insignificant (*Cacciatore et al., 2019*). The ANB angle increased more effectively over a short period of time in the treatment of skeletal class III malocclusion using maxillary protraction during the growth period, compared to that in the untreated group; however, the effect become less pronounced toward the end of growth period and the skeletal pattern became almost similar to that of the untreated control group (*Vaughn et al., 2005*; *Mandall et al., 2010*, *2012*, *2016*). These reports suggest that the skeletal pattern is not significantly altered by orthodontic treatment or residual growth at least in participants

without CLP or ICP. However, we selected younger patients in Hellman's stage IIA before they underwent orthodontic intervention because we wished to exclude the effects of orthodontic treatment on the skeletal pattern and thus on the determination of typical morphological characteristics of ICP, which are encountered at the beginning of the orthodontic treatment. Further studies with samples after growth complete will be required for better evaluation and understanding of craniofacial morphology of ICP.

The present study population included patients with clefts of the soft palate and those with clefts of the hard and soft palate. Surgeries for clefts of the soft palate are often less invasive because they require lesser removal of mucosa from the nasal side than that required for clefts of the hard and soft palate. Thus, this research was also aimed at confirming whether the differences in the cleft type affect the degree of maxillary growth suppression. Moreover, the differences between the skills of the four surgeons were also considered because they may affect maxillary development. Table 3 shows the sample distribution for each category. Fisher's exact test was used to examine the distribution of the small sample size. The data in Tables 3A and 3B exhibit *P*-values greater than 0.05 and relatively high power $(1-\beta)$, indicating that the distribution was not significantly different. These results suggest that differences in the cleft type (cleft range) and skills of the operating surgeon were not significantly associated with the craniofacial morphology of patients with ICP. Therefore, the characteristics of three clusters obtained from this ICP population can be expected to represent the typical craniofacial morphologies encountered in the clinical practice, which reflect the characteristics of ICP itself and the effects of any subsequent surgeries. This information could be beneficial in orthodontic treatment planning for patients with cleft palate.

## CONCLUSIONS

Cluster analysis of craniofacial morphologies in patients with ICP resulted in the identification of three clusters: a cluster with a relatively harmonious anteroposterior relationship between the maxilla and the mandible; that with anterior crossbite caused by a significantly small maxilla; and that with a smaller mandible with posterior rotation showing a skeletal class II relationship. A total of 44.4% of the patients were classified into the cluster characterised by a skeletal class II relationship with a smaller mandible (cluster C). Differences in the cleft type (cleft range) and skills of the surgeon were not associated with the distribution of craniofacial morphology. These results suggest that the mechanisms responsible for craniofacial morphology of patients with ICP are different from those in patients with CLP. Understanding the typical morphological characteristics could enable better diagnostic categorisation of patients with ICP, which may eventually improve orthodontic treatment planning.

## ACKNOWLEDGEMENTS

We wish to thank all the members of our department for participating in useful discussions on the topic of the study.

### Funding

This work was supported by JSPS KAKENHI Grant Number JP20K10228. The funders had no role in study design, data collection and analysis, decision to publish, or preparation of the manuscript.

### Grant Disclosures

The following grant information was disclosed by the authors:
JSPS KAKENHI: JP20K10228.

### Competing Interests

Naoto Haruyama is an Academic Editor for PeerJ.

### Author Contributions

- Kyoko Tsuji conceived and designed the experiments, performed the experiments, analyzed the data, prepared figures and/or tables, authored or reviewed drafts of the paper, and approved the final draft.
- Naoto Haruyama conceived and designed the experiments, performed the experiments, analyzed the data, prepared figures and/or tables, authored or reviewed drafts of the paper, and approved the final draft.
- Shunsuke Nomura analyzed the data, prepared figures and/or tables, and approved the final draft.
- Naohisa Murata analyzed the data, prepared figures and/or tables, and approved the final draft.
- Keigo Yoshizaki analyzed the data, prepared figures and/or tables, and approved the final draft.
- Takeshi Mitsuyasu performed the experiments, authored or reviewed drafts of the paper, and approved the final draft.
- Hiroyuki Nakano performed the experiments, authored or reviewed drafts of the paper, and approved the final draft.
- Seiji Nakamura conceived and designed the experiments, authored or reviewed drafts of the paper, and approved the final draft.
- Yoshihide Mori conceived and designed the experiments, authored or reviewed drafts of the paper, and approved the final draft.
- Ichiro Takahashi conceived and designed the experiments, authored or reviewed drafts of the paper, and approved the final draft.

### Human Ethics

The following information was supplied relating to ethical approvals (i.e., approving body and any reference numbers):

This study protocol was reviewed and approved by the Kyushu University Institutional Review Board for Clinical Research (#27-135).

## Data Availability

The raw measurements are available in the Supplementary Files.

## Supplemental Information

Supplemental information for this article can be found online at http://dx.doi.org/10.7717/peerj.11297#supplemental-information.

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
