# Peer review of "Characteristics of craniofacial morphology and factors affecting them in patients with isolated cleft palate"

_PeerJ, doi:10.7717/peerj.11297_

## Round 0.1 · original submission · Minor Revisions

Basically, the three reviewers are positive about the methodology of your study, but they have a number of suggestions to improve the reporting. I would like to point to a few:

1. The introduction could be better focused. It lasts until line 98 before the reader gets the message that this paper is about isolated cleft palate and not about cleft lip and palate. There are also unsubstantiated claims in this section like Furlow would be amongst the most widely used techniques?

2. The M&M section lacks important details. The study heavily relies on cluster analysis. As all referees pointed out, for the cluster analysis you refer to a paper that is not accessible (Yamanouchi 1995), therefore the cluster analysis should be explained extensively in the M&M section. Please make sub-paragraphs in the M&M section: sample, cephalometric analysis, statistics.

3. The sample lacks a detailed description for the timing of surgery. Were patients with a Pierre Robins Sequence excluded? Apparently, the authors did not perform a power calculation. I would like to see at least a post-hoc power calculation to estimate if the study was sufficiently powered.

For all other comments, I refer to the referee reports.

·

Basic reporting

I commend the authors for their study. The manuscript is clearly written in professional, unambiguous language and highlights the variability of craniofacial morphology found in patients with isolated cleft palate.
I have listed a few items that require clarification in the experimental design section of this review (see below) which I feel if addressed will add strength to their paper

Experimental design

The selection of patients (36 females aged 5.36 years) requires some clarification. These children were selected over a 12 year period (approx. 3 per year) but there is no indication of what proportion of total population pool of children with ICP these selected children represent.

In addition it appears that all these girls required orthodontic intervention at 5.36 years of age (hence the need for a lateral cephalogram) This implies the sample chosen may not be representative of all the ICP population as some will not be of need for orthodontic intervention at this age.

The authors state in 141-143 that linear measurements were normalised based on UAFH and reference Yamanouchi -unfortunately I could access this reference (I suspect it is Japanese) but perhaps the authors could justify this approach - one would assume that the standardised ruler in the head positioning device on the lateral cephalostat would have enabled calibration of the lat cephalograms for all linear measurements

The authors also mention in the discussion (lines 253) that normalisation of the linear measurements was achieved by dividing the values with the anterior facial height (N-ANS), as this is a reliable measurements and remains considerably unaffected by age or growth’ - this may require further clarification as I am unsure why this was required if the linear measurements were taken from calibrated lateral cephalograms

Validity of the findings

The authors compare their lateral cephalometric findings to those of Japanese norms but it is unclear if these norms are for female at this age or whether the norms used were from combined male and female subjects

Additional comments

The authors should be congratulated in collecting the study sample, and for their cluster analysis. Clarification on sample selection and methods for calibrating the lat cephalogram linear measurements would strengthen this paper

·

Basic reporting

I would like to thank the authors for a well written report. My suggestions are:

Introduction:
You reference Wardill and Furlow as most widely used techniques, but where and by who as these are not widely used in some countries at all

Experimental design

Suggestions:

Not clear if this a retrospecitve or prospective cohort study sample
Please explain Hellman's stage IIA for reader
Cephalogram taken on same machine to account for magnification factor?
Were the profilograms used gender specific in view of all female sample?
Figures 1 and 2 relate to Method but are not mentioned in that section
this particularly relates to method of clustering which needs clarifying
Does figure 1 need every point measured explaining?

Validity of the findings

Discussion

Suggest a section on strengths and weaknesses of the approach including gender bias, small sample, age related findings before growth complete but before intervention

Reviewer 3 ·

Basic reporting

The authors have made a great effort in reporting their data in ICP patients. This is usually a smaller group in cleft patients and the majority craniofacial morphology is about cleft lip, cleft lip alveolus and UCLAP patients. It is interesting to see which clusters come from the analysis though I found it a difficult point of view in the beginning.

legends and tables ok. raw data shared.

the structure is sometimes confusing. The focus is not always related to the aim:
The fact that this study is about ICP should be “ sold” more. ICP and growth is first mentioned in line 99 of the introduction. That is a shame, as a reader I would like to see the importance and focus on ICP sooner. Some better focus in de the discussion would also help readability, because the focus on pierre robin is confusing. Isn’t it more logical to put line 291-296 (adapted) before the pierre robin explanation?

Experimental design

It is always a challenge to get numbers in cleft groups:Cleft palate patients appr 5y of age were taken and only girls were included, because the group of boys was too small. Do the authors expect a lot of difference in growth between boys and girls at that age? would the group be much bigger with boys included? Considering that 4 surgeons operated this group of patients, which may have an effect on the results
I unfortunately had no access to the Japanese growth study but I presume the mean numbers at the age of 5 were taken for only girls?
I found the cluster analysis difficult in the beginnen to understand and unfortunately could not access the study of Yamanouchi et al. is there a possibility for another reference?
Amount of surgeons were taken into account (maybe mention in discussion something about effect surgeon)
Measurement error: only for distances not angles.

Validity of the findings

It is nice that clusters could be found, yet the groups are rather small. Do the authors have confidence intervals next to p-value

Additional comments

please look at the structure of the article and how this is related to the findings. It will help to keep the reader focused on the aim and the results.
the problem of course is the smaller groups within the group and 4 surgeons, even though no difference was found. may be some extra explanation on this limit should be added in the discussion

---

## Round 0.2 · Minor Revisions

Thank you for the revisions you made and previous comments have been dealt with satisfactorily. There are only a few minor comments left, see reviewer 1, but these can be easily addressed.

I still would like to see also a better description of the inclusion/exclusion criteria of the patient group. It has been clarified in the revised version but it is still not completely clear. Please make a clear statement like:
Inclusion criteria were: patients with overt ICP, no syndromes affecting the craniofacial morphology. Exclusion criteria were: males, syndromes affecting the craniofacial morphology including the Pierre Robin sequence, submucous cleft palate, missing diagnostic records??. I wrote this just as an example, I don't know if it is correct.

You state that n=77 patients with ICP were identified during the recruiting period. 36 girls and 12 boys. That means n=29 were excluded for other reasons but this is not mentioned. Please give numbers with reasons for exclusion.

If you could address these last comments, the paper will be considered for acceptance.

·

Basic reporting

no comment

Experimental design

no comment

Validity of the findings

no comment

Additional comments

Characteristics of craniofacial morphology and factors affecting them in patients with isolated cleft
palate (#52363-v1)
General comments:
Thank you for undertaking the changes previously recommended.
I am satisfied with these changes and suggest that this manuscript be accepted for publication with minor edits noted below:
Minor edits
The text which has been added on lines 136-160 – already appears within the Figure 1 legend and this duplicated text can be deleted from this section of the paper.
. N: nasion, the anterior-most point of the frontonasal suture; S: sella turcica, the estimated centre of the hypophyseal fossa; Or: orbitale, the deepest point on the infraorbital margin; Po: porion, the upper margin of the ear canal; Ba: basion, the anterior-most margin of the foramen magnum; Cd; condylion, the most supradorsal point of the condylar head; Ar: articulare, the intersection between the external contour of the cranial base and dorsal contour of the condylar head and neck; Ptm: pterygomaxillary fissure, the inferior point of the fissure; PNS: posterior nasal spine; ANS: anterior nasal spine; A: subspinale, the deepest point on the premaxilla between the ANS and prosthion in the midline; U1 (Is) and U1a : upper incisor constructed between the incisal tip of the anterior-most deciduous maxillary central incisor and its apex; Mo: mid-point of the deciduous maxillary second molar; L1 (Ii) and L1a: lower incisor constructed between the incisal tip of the most anterior deciduous mandibular central incisor and its apex; Pog: pogonion, the anterior-most point of the bony chin; B: supramentale, the posterior-most point in the concavity between the infradentale and Pog; Me: menton, the lowest point on the symphyseal shadow; Gn: gnathion, the point on the chin determined by bisecting the angle formed by the facial plane and mandibular plane; Go: gonion, the intersection between the ramus plane and the mandibular plane; A’: intersection of a perpendicular drawn from point A to the palatal plane; Ptm’: intersection of a perpendicular line from the Ptm to the palatal plane; Ms’: intersection of the perpendicular drawn from Mo to the palatal plane; U1 (Is): upper incisor constructed between the incisal tip of the most anteriorly placed deciduous maxillary central incisor; Is’: intersection of the perpendicular drawn from Is to the palatal plane; L1 (Ii): lower incisor constructed between the incisal tip of the most anteriorly placed deciduous mandibular central incisor; Ii’: intersection of the perpendicular drawn from Ii to the palatal plane; Mo: mid-point of the deciduous maxillary second molar; Ms’: intersection of the perpendicular drawn from Mo to the palatal plane; Mi’: intersection of the perpendicular drawn from Mo to the mandibular plane.
Line 172 currently states “…. size of the maxilla [point A-pterygomaxillary fissure (A’-Ptm’)], size of the mandible [condylion-gonion (Cd-Gn)],… “ the term ‘length’ would be a more appropriate descriptor of what has been measured than the term ‘size’. …likewise on line 268
Line 220 currently states “…. exhibited a significantly larger Gn-Cd (mandibular length) and A-B plane angle than the standard…” the term ‘longer’ would be a more appropriate descriptor when referencing Gn-Cd (mandibular length) than the term ‘larger’.

·

Basic reporting

Previous comments have ben dealt with satisfactorily

Experimental design

Previous comments have ben dealt with satisfactorily

Validity of the findings

Previous comments have ben dealt with satisfactorily

Additional comments

Previous comments have ben dealt with satisfactorily

---

## Round 0.3 · accepted · Accept

I looked over the last minor changes and all have been addressed adequately. Thank you.